# RNA G-Quadruplex within the 5′-UTR of FEN1 Regulates mRNA Stability under Oxidative Stress

**DOI:** 10.3390/antiox12020276

**Published:** 2023-01-26

**Authors:** Ying Ma, Yang Yang, Jingyu Xin, Lingfeng He, Zhigang Hu, Tao Gao, Feiyan Pan, Zhigang Guo

**Affiliations:** 1Jiangsu Key Laboratory for Molecular and Medical Biotechnology, College of Life Sciences, Nanjing Normal University, Nanjing 210023, China; 2College of Life Science, Northeast Agricultural University, Harbin 150030, China

**Keywords:** DNA base excision repair (BER), reactive oxygen species (ROS), oxidative stress, G-quadruplex, FEN1, hnRNPA1

## Abstract

Reactive oxygen species (ROS) are a group of highly oxidative molecules that induce DNA damage, affecting DNA damage response (DDR) and gene expression. It is now recognized that DNA base excision repair (BER) is one of the important pathways responsible for sensing oxidative stress to eliminate DNA damage, in which FEN1 plays an important role in this process. However, the regulation of FEN1 under oxidative stress is still unclear. Here, we identified a novel RNA G-quadruplex (rG4) sequence in the 5′untranslated region (5′UTR) of FEN1 mRNA. Under oxidative stress, the G bases in the G4-forming sequence can be oxidized by ROS, resulting in structural disruption of the G-quadruplex. ROS or TMPyP4, a G4-structural ligand, disrupted the formation of G4 structure and affected the expression of FEN1. Furthermore, pull-down experiments identified a novel FEN1 rG4-binding protein, heterogeneous nuclear ribonucleoprotein A1 (hnRNPA1), and cellular studies have shown that hnRNPA1 plays an important role in regulating FEN1 expression. This work demonstrates that rG4 acts as a ROS sensor in the 5′UTR of FEN1 mRNA. Taken together, these results suggest a novel role for rG4 in translational control under oxidative stress.

## 1. Introduction

G-rich sequences are significantly enriched in the gene promoter region [1]. It is well known that repeated G bases can fold into DNA G-quadruplex (dG4) and RNA G-quadruplex (rG4) structures, a four-chain secondary structure, stabilized by Hoogsteen-Hydrogen bonds under physiological conditions [2]. In eukaryotes, DNA mostly exists in the nucleus in a classic double-stranded helical conformation, while RNA is single-stranded, located both in the nucleus and in the cytoplasm. Therefore, in general, RNAs have greater freedom to fold into different secondary structures, including rG4s, which play key regulatory roles in biological processes including translation [3]. rG4s located in the 5′UTR and 3′UTR region play an important role in promoting or repressing the translation of various proteins [4]. Changes in the physical and chemical environment lead to instability of G4, affecting gene expression and the occurrence of certain diseases [5,6]. Notably, the formation of G4 may suppress the expression of some cancer-related genes [7,8]. The DNA oxidation caused by reactive oxygen species (ROS) is one of the key factors inducing DNA damage, causing direct changes to the structure and the coding sequence of the DNA, thus affecting gene expression [9]. It is known that oxidative damage occurs most frequently at guanine (G), as it has a lower redox potential than the other nucleobases [10]. ROS attack G directly to form 8-oxo-7,8-dihydroguanine(8-oxoG) [11]. The presence of 8-oxoG in potential quadruplex sequences (PQSs) may affect G4’s structure and gene expression [12].

To repair the DNA damages caused by oxidative stress, cells have evolved the base excision repair (BER) system, which needs to be precisely regulated to ensure the effective repair [13,14]. The molecule 8-oxoguanine DNA glycosylase 1 (OGG1) can be regulated by anti-oxidative transcription factors NRF-1 and NRF-2α [15]. Apurinic/apyrimidinic endonuclease 1 (APE1) is phosphorylated upon oxidative stress, which inhibits its endonuclease activity [16,17]. DNA glycosylase Ntg1 can be translocated to mitochondria from the nucleus, helping to eliminate oxidative DNA damage in mitochondria [18]. In the BER system, Flap endonuclease 1 (FEN1) is the key enzyme involved, whose mutation causes defects in RNA primer removal and long-patch BER, resulting in numerous DNA breaks [19]. FEN1 is highly expressed in several human cancer cells and associated with an increased tumor grade and aggressiveness [20]. In addition, the overexpression and hypomethylation of the FEN1 promoter may serve as biomarkers for breast cancer progression, which is accompanied by a high level of oxidative stress [21,22,23]. Our previous studies have revealed that post-transcriptional modifications (PTMs) are essential for FEN1 function in DNA replication and cell cycle progression. Notably, several lines of evidence show that FEN1 plays an important role in response to oxidative stress. FEN1 is localized in mitochondria, where it is proposed to scavenge ROS [24,25,26]. FEN1 can be recruited to DNA oxidative damage sites under oxidative stress [27]. A FEN1-deficient cell is hypersensitive to H_2_O_2_. Although these researches have demonstrated the key role of FEN1 under oxidative stress, the expression of FEN1 under oxidative stress is unknown.

In the present study, we found that FEN1 was significantly downregulated under oxidative stress and this downregulation was due to the instability of FEN1 mRNA. After a series of biological and chemical analyses, our data revealed that the presence of an rG4 structure in the 5′UTR of FEN1 mRNA and its recognition by hnRNPA1 was indispensable for the stability of FEN1 mRNA. Our finding suggests the regulatory function of rG4 in FEN1 expression and may provide a broader understanding of how the BER system responds to oxidative stress through the rG4 oxidation-mediated regulation of FEN1.

## 2. Materials and Methods

### 2.1. Cell Culture and Treatment

MCF-7, SW480, HCT116, NCI-H460, A549 were purchased from ATCC. MCF-7 and SW480 were grown in Dulbecco’s modified Eagle’s medium (DMEM) containing 10% FBS, 10 unit/mL penicillin and 10 g/mL streptomycin. HCT116, NCI-H460 and A549 were cultured in Roswell Park Memorial Institute (RPMI)-1640 medium. Cells were sub-cultured and seeded in 6-well plates for treatment of H_2_O_2_ upon reaching 80% confluence. After the cells were treated with H_2_O_2_ and allowed to recover for a certain period of time, the cells were harvested for RT-qPCR or Western blot analysis.

### 2.2. Antibody and Beads

The hnRNPA1 (Cat# A7491), YY1 (Cat# A19569), β-tubulin (Cat# A14991), Lamin B1 (Cat# A16909) antibodies and secondary antibodies were purchased from ABclonal (Wuhan, China) and the FEN1 antibody (Cat# GTX101777) was obtained from GeneTex (Irvine, CA, USA). Phospho-Histone H2A.X (Ser139) antibody (γH2AX) (Cat# 2577) was ordered from Cell Signaling Technology (Danvers, MA, USA). The DNA/RNA oligonucleotides and modified sequences are listed in the Appendix A and were all purchased from Sangon Biotech, (Shanghai, China). Dynabeads M280 streptavidin magnetic beads were obtained from Thermo Fisher Scientific (Waltham, MA, USA).

### 2.3. Real-Time Quantitative PCR (qPCR) Analysis

Total cellular mRNA was extracted with the RNA isolator Total RNA Extraction Reagent (R401-01, Vazyme, Nanjing, China) according to the manufacturer’s instructions. Reverse transcription into cDNA was performed using an RT reverse transcription kit (R122-01, Vazyme, Nanjing, China). qPCR was performed using a 20 μL reaction mix containing cDNA, AceQ qPCR SYBR Green Master Mix (High ROX Premixed) (Q141-02, Vazyme, Nanjing, China) and primers. Each PCR amplification was performed at least 3 times and repeated in three independent experiments. The primers used for FEN1 and β-actin in this study are listed in Appendix A.

### 2.4. Western Blotting

Cellular protein samples were obtained by lysing cells with 1% SDS lysis buffer supplemented with 1 mM PMSF. A volume of each cell sample was loaded on SDS-polyacrylamide gel electrophoresis (SDS-PAGE) at 85 V for 40 min, then 120 V for 1–2 h, using the eBlot™ L1 Fast Transfer Membrane Transfer System (L00686C, Genscript, Nanjing, China) for membrane transfer. Membranes were then blocked with 5% nonfat dry milk for 2 h. After blocking, membranes were incubated with the indicated primary antibodies overnight at 4 °C and washed 3 times with PBST. Membranes were then incubated with the corresponding secondary antibodies against goat anti-rabbit IgG (H + L) HRP (BS13278, Bioworld, Nanjing, China) and goat anti-mouse IgG (H + L) HRP (BS12478, Bioworld, Nanjing, China) for 2 h at room temperature and washed again with PBST. Membranes were analyzed, scanned with a Tanon 4500 imaging system (Tanon, Shanghai, China), and quantified with ImageJ software (National Institutes of Health, Bethesda, MD, USA).

### 2.5. Circular Dichroism (CD) Analysis

Synthesized oligonucleotides of PQSs in the FEN1 5′UTR (containing mutations, Appendix A) were diluted to 5 μM in G4 buffer (50 mM Tris-HCl, pH 7.4, 100 mM KCl and 12 mM NaCl). After heating at 95 °C for 5 min, the samples were slowly cooled to room temperature at a rate of 0.01 °C/s. Next, the CD spectra were recorded on a Circular Dichroism Spectropolarimeter (Chirascan, Applied Photophysics, Surrey, UK) from 220 to 320 nm, with a 1 mm path length and a 1 nm bandwidth. The spectra are reported as ellipticity (mdeg) versus wavelength (nm). The spectra for all samples were baseline-corrected with buffer and represent the average of three runs.

### 2.6. Native Polyacrylamide Gel Analysis

Commercially synthesized wild-type and mutated type 5′-FAM-labeled RNA were diluted to 5 μM in G4 buffer. The samples were heated to 95 °C for 5 min and then slowly cooled to room temperature at a rate of 0.01 °C/s. Then, the samples were loaded on 12% native-PAGE gels containing 100 mM KCl and electrophoresed at 4 °C at a rate of 8 V/cm; the gel was then scanned using an Amersham Imager 600 (GE Healthcare, Chicago, IL, USA).

### 2.7. Pull-Down Assay and Liquid Chromatography-Tandem Mass Spectrometry (LC-MS/MS)

A total of 10 μM biotinylated oligonucleotides was folded in the above G4 buffer. MCF-7 cell nuclear lysates were isolated by a nuclear and cytoplasmic extraction kit (CW0199, CWBIO, Beijing, China). The nuclear protein concentration was adjusted to 1 mg/mL. The rG4–biotin and lysate were allowed to react for 3 h at 4 °C in RNA pull-down buffer (50 mM Tris–HCl, pH 8.0, 100 mM KCl, 20% (*v/v*) glycerol, 0.2 mM ethylenediaminetetraacetic acid (EDTA), 1 mM DTT, 0.01% Nonidet-P40, 50 μg/mL yeast tRNA (Ambion) and 160 U/mL RNasin). Dynabeads^TM^ M-280 streptavidin beads (11205D, Invitrogen, Carlsbad, CA, USA) were incubated with the rG4–protein complex at 4 °C for 2 h, washed three times with RNA wash buffer (10 mM Tris–HCl, pH 8.0, 100 mM KCl, 20% (*v/v*) glycerol, 0.2 mM ethylenediaminetetraacetic acid (EDTA), 1 mM DTT and 0.01% Nonidet-P40), for 5 min each time at 4 °C. The reactions were quenched by adding 40 μL of 2× loading buffer, heated at 95 °C for 5 min and resolved by SDS-PAGE. Affinity-captured proteins were digested with trypsin and then analyzed by LC-MS/MS.

### 2.8. Electrophoresis Mobility Shift Assay (EMSA)

The 5′-FAM-labeled RNA was diluted to 5 μM in G4 buffer. After heating at 95 °C for 5 min and slowly cooling to 25 °C at a rate of 0.01 °C/s, various amounts of purified His-tagged hnRNPA1 (BSA as the control) were incubated with rG4 in binding buffer (100 mM Tris–HCl, 100 mM NaCl, 10 mM MgCl_2_, 10% (*v/v*) glycerol and 100 mM KCl). The binding reaction was performed at 4 °C for 20 min. Then, the samples were loaded on a 12% native-PAGE gel containing 100 mM KCl and electrophoresed at 4 °C at a rate of 8 V/cm, and the gels were imaged using an Amersham Imager 600 (GE Healthcare, Chicago, IL, USA).

### 2.9. Immunofluorescence

MCF-7 cells were fixed in 4% paraformaldehyde (PFA) for 30 min at room temperature and permeabilized with 0.5% Triton X-100 (*v/v*) in PBS. For DNA damage detection, the cells were treated with γH2AX antibody (1:200 dilution, #9718, Cell Signaling Technology) after blocking with 3% BSA. The cells were washed three times with PBST and incubated with goat anti-rabbit IgG (H + L) highly cross-adsorbed secondary antibody, tagged with Alexa Fluor Plus 488 for 1 h at 37 °C, then washed three times with PBST. After washing, the cells were incubated for 10 min with 4′,6-diamidino-2-phenylindole (DAPI) for nucleus staining. The cells were washed three times with PBS and mounted in 1,4-diazabicyclooctane triethylenediamine. The fluorescence was observed by confocal microscopy (MRC-1024, BIO-RAD, Hercules, CA, USA).

### 2.10. Statistical Analysis

The results are presented as the means ± SEM. Data from three independent experiments were subjected to statistical analysis using Prism 6 software (GraphPad Software, La Jolla, CA, USA). *p*-values were calculated using the Student’s *t*-test. Differences with *p* < 0.05 were considered significant.

## 3. Results

### 3.1. FEN1 Is Downregulated under Oxidative Stress

To evaluate the effects of oxidative agents on FEN1 expression, cells were treated with 300 μM H_2_O_2_ for 2 h and recovered for different times. Unexpectedly, the protein level of FEN1 decreased gradually with the recovery time (Figure 1A,B), which was significant at the 4 h recovery time point. Therefore, 300 μM H_2_O_2_ treatment for 2 h and then recovery for 4 h was applied in the follow-up experiments. To confirm this result, we examined the mRNA and protein levels of FEN1 in other cancer cells with the same treatment (Figure 1C–E). The results further proved that FEN1 was downregulated by oxidative stress. Since YY1 is a transcriptional repressor of FEN1, we first tested the expression of YY1 under oxidative stress and no significant change was observed (Figure 1F,G). The nucleocytoplasmic separation experiment further showed that the ratio of YY1 in the nucleus was not increased significantly (Figure 1H,I). These results suggested that there may be other regulatory mechanisms for the downregulation of FEN1 under oxidative stress. 

### 3.2. G-Quadruplex Structure Exists in the 5′UTR Region of FEN1 mRNA

The secondary structure of nucleic acids affects gene expression, localization and function [28]. The G-quadruplex is one kind of such structure whose oxidation plays an important role in gene expression. By employing bioinformatic analysis tools (http://bioinformatics.ramapo.edu/QGRS/analyze.php, accessed on 15 January 2020), six PQSs were found in the FEN1 5′UTR region (+55 to +108 nt, named r-PQS1-6) (Figure 2A,B). To confirm whether the predicted PQSs form G4 structures or not, the oligonucleotides of six PQSs were synthesized (Appendix A). The oligonucleotides were thermally annealed in G4 buffer, and subjected to CD spectra analysis. Among the six PQSs tested, r-PQS4 (abbreviated as rG4 in the following) showed a negative peak at 245 nm and a positive peak at 265 nm, indicating a typical parallel G4 structure (Figure 2C). To verify whether rG4 conforms to the structural features of G4, we first tested the influence of different metal ions (such as K^+^ and Na^+^) on the formation of G4. The results showed that K^+^ strongly increases the formation of the parallel G4 structure (Figure 2D). Then, we monitored the CD spectra of rG4 in the temperature range of 10~90 °C to confirm the stability of the formed G4 structure (Figure 2E,F). 

Since G-quadruplex structures are mostly composed of three or more G-tracks, the number and arrangement direction of the G-tracks are different, and the types of G-quadruplex structures are also different, which affects the formation of the G-quadruplex [29]. To explore which G-track is important for the formation of rG4, we replaced guanine (G) with adenine (A) in the four G-tracks, respectively, to obtain four mutants (rG4m1~4) (Figure 2G, Appendix A). CD spectroscopy results showed that the molar ellipticity of rG4m4 (G→A, at +103~+106 nt) at 265 nm was reduced by four times compared with rG4 (Figure 2H,I), suggesting that the fourth G-track played a key role in the formation of the G-quadruplex structure, which can be used for subsequent experimental studies.

It is well-known that G is oxidized to 8-oxoG by a variety of ROS. To investigate whether the rG4 structure was affected by oxidative stress, we analyzed the rG4 structure treated with different concentrations of H_2_O_2_. CD spectroscopy results showed that the molar ellipticity of rG4 at 245 and 265 nm was decreased with the increased concentration of H_2_O_2_ (Figure 2J,K). Furthermore, to evaluate the importance of rG4m4 oxidation to the G4 structure, we synthesized a “mutation” rG4 where the G at +104 position was replaced with 8-oxoG. As shown in CD spectrum analysis in Figure 2L, the G4 structure was significantly disrupted in the mutant chain. Together, these data suggested that FEN1 mRNA could form rG4 structure which was destabilized by oxidative stress. 

### 3.3. TMPyP4 Affects rG4 Formation

To further explore the effect of the G4 structure on FEN1 mRNA, we used TMPyP4, a G4-specific ligand, in our next experiments. UV-Vis spectroscopy was used to test the interaction of rG4 with TMPyP4. We observed a large shift in the absorption spectrum, when TMPyP4 was added to the pre-folded rG4 solution (Figure 3A). This was due to the transfer of the p-electron from the purine base to the pyrrole ring, which suggested that TMPyP4 was stacked on rG4, or possibly intercalated between G-tetrads. To examine as to how TMPyP4 can interact with FEN1 rG4, we added different concentrations of TMPyP4 to single-stranded RNA solution before annealing, then performed CD spectroscopy (Figure 3B,C). Results showed that 10 μM TMPyP4 significantly reduced the molar ellipticity at 265 nm, the negative peak at 245 nm disappeared, and the secondary structure was hardly formed at the concentration of 100 μM, indicating that as the concentration of TMPyP4 increased, more unfolded forms of rG4 were observed. This result was also confirmed by native PAGE electrophoresis, as shown in Figure 3D,E. With the increase of TMPyP4 concentration, more slowly migrated bands appeared in the gel, suggesting more unfolded single-stranded RNA. Furthermore, rG4 formed from single-stranded nucleotides migrated faster on native PAGE gels than single-stranded nucleotides, possibly due to their more compact structure. As shown in Figure 3F, annealing had no effect on the migration of RNA strands on the gel. Next, to explore the effect of rG4 on FEN1 expression, we pretreated cells with TMPyP4 for 24 h before cell lysis. Our data showed that with increasing concentration of TMPyP4, the expression of FEN1 gradually decreased (Figure 3G–I). Thus, TMPyP4 can bind to FEN1 RNA and inhibit rG4 formation in a dependent manner. To further clarify the importance of rG4 to FEN1 expression, pyridostatin (PDS), a stabilizer of the rG4 structure, was used in the experiment. The result showed that the expression of FEN1 was upregulated under oxidative stress with PDS pretreatment (Figure 3J–O). 

### 3.4. FEN1 rG4 Interacts with hnRNPA1

To understand the mechanism by which rG4 affects FEN1 expression, we first focused on the rG4 interacting proteins. Biotin-labeled rG4 oligonucleotides were incubated with nuclear extract, and pulled down by magnetic streptavidin beads. The precipitated proteins were then subjected to SDS-PAGE and mass spectrum analysis (Figure 4A). The MS-obtained differential binding proteins were subjected to functional enrichment analysis using WebGestalt (http://www.webgestalt.org/, accessed on 10 August 2020). As shown in the Figure 4B,C, the proteins in both the control group and the FEN1 rG4 group were enriched in ribosome biogenesis and RNA metabolism. Among these proteins, heterogeneous nuclear ribonucleoproteins in RNA-binding proteins (hnRNPA1) exhibited enhanced binding to rG4 (Figure 4D). We first confirmed the MS result by a pull-down assay using purified recombinant human hnRNPA1 protein and biotin–rG4 (Figure 4E,F). Then, we predicted protein–RNA binding probabilities using online software (http://pridb.gdcb.iastate.edu/RPISeq/index.html, accessed on 13 August 2020). The calculated data showed that the interaction probability between the hnRNPA1 RNA-binding domain and FEN1 rG4 is 0.7, indicating a strong binding. Furthermore, this interaction was verified by an EMSA assay (Figure 4G). To further confirm the combination in the cells, rG4 labeled with FAM was transfected into the cells. Immunofluorescence data showed the co-localization of rG4 and hnRNPA1 in cells (Figure 4H). These results indicated that hnRNPA1 can specifically bind to the rG4 of FEN1.

### 3.5. HnRNPA1 Regulates FEN1 Expression by Affecting rG4 Formation

After proving specific interaction between hnRNPA1 and FEN1 rG4, the correlations between the expression of FEN1 and hnRNPA1 in different types of cancer tissues were analyzed by the GEPIA database. Figure 5A–E showed that in various tumors, the correlation coefficient of FEN1 and hnRNPA1 expression was greater than zero, indicating that the expression of these two genes is positively correlated in tumors. Then, we speculated whether hnRNPA1 could regulate FEN1 expression through such an interaction. To test this hypothesis, we applied Western blot to examine FEN1 expression in cells with or without hnRNPA1 knockdown (Figure 5F,G). The results showed that the protein level of FEN1 was significantly decreased in hnRNPA1-knockdown cells. Considering FEN1 plays an important role in DNA repair under oxidative stress, we then wondered whether hnRNPA1 might be involved in DNA repair through regulating FEN1 expression. Immunofluorescence staining of γH2AX (a well-known DNA damage marker) confirmed our speculation, because the foci numbers of γH2AX were significantly increased with hnRNPA1 knockdown, suggesting more DNA damage was accumulated and DNA repair was impaired (Figure 5H,I). 

## 4. Discussion

Under oxidative stress, ROS continues to invade the DNA structure, leading to oxidation and cleavage of nucleobases [30]. It has been increasingly recognized that the gene expression depends in part on the regulation of oxidative DNA damage and repair. BER is the primary pathway that is critical for removing oxidative DNA damage, and thus maintaining genomic integrity. FEN1 is involved in the excision of DNA and RNA substrates during BER and plays important roles in multiple DNA metabolic pathways. Therefore, the precise regulation of FEN1 is the key to maintaining genomic stability. Our previous studies also showed that under oxidative stress, FEN1 can be recruited by H4R3me2s to DNA oxidative damage sites to enhance its nuclease activity and BER efficiency. It was also found that FEN1 was upregulated in HeLa cells under oxidative stress [27]. However, in this study, we found that FEN1 was downregulated in A549, NCI-H460, MCF-7, SW480, and HCT116 cells under oxidative stress, which indicated that the regulation of FEN1 expression might be inconsistent in different cells and tissues. Further, increasing evidence suggests that oxidative stress preferentially targets G4-forming sequences [31]. Oxidative damage affects the folding and unfolding of the G4 structure, which may play a regulatory role in gene expression.

The classical sequence motif of G4 is usually denoted as 5′-G ≥ 3NXG ≥ 3NXG ≥ 3NXG ≥ 3-3′ (x = 1–7 nucleotides), and this sequence pattern has been used to identify PQSs in human genomes through bioinformatics analysis [32,33,34]. More than 40% of human gene promoters contain one or more potential G4 elements, such as *KIT* [35] and, especially in some oncogenes, *MYC* [36]. In addition, the 5′UTR in approximately 3000 genes contains rG4 [37]. rG4 is considered to be an important player in biological and biomedical events and related to the regulation of various functions [38]. Many studies have focused on the effects of G4 oxidation in the regulatory regions of the genome, and the regulation of RNA metabolism by rG4 [39,40]. Here, we found that the 5′UTR region of FEN1 mRNA contains a number of G4 motifs. The newly discovered rG4 may have potential effects on the stability of FEN1 mRNA. Recently, studies have found that the average density of PQS in human DNA repair genes is nearly two times greater than the average density of PQS in all coding and non-coding human genes [41]. Research highlights the link between human DNA repair genes and the G4 sequence, but the mechanism by which G4 affects gene expression in the context of this complex structure in cells is unclear. Our study identified potential G4 structures in the 5′UTR of FEN1 mRNA and found that the +71~+108 sequences can form a typical parallel G-quadruplex structure. We further revealed the association of rG4 structure with FEN1 expression under oxidative damage conditions. Therefore, FEN1 rG4 in this study may pave the way for understanding how these DNA repair-related genes are regulated by PQS under oxidative stress.

The important functional significance of oxidative base modifications in the G4-forming sequence is their effect on G4 protein interactions. When G is oxidized to 8-oxoG or other products; this site cannot participate in G: G Hoogsteen base-pairing in the tetrad, resulting in disruption of the G4 structure [42]. Previous studies have shown that hnRNPA1 specifically recognizes telomere G4 [43]. Tatsuya Nishikawa et al. also reported that hnRNPA1 interacts with the G-quadruplex in the TRA2B promoter [44]. With the discovery of FEN1 rG4, it is of great importance to find its interacting proteins for studying the regulation of FEN1 expression. Here, we have found that the sequence corresponding to the rG4 structure binds with the hnRNPA1 protein in vitro and in vivo. Our study expands the understanding of FEN1 regulation by oxidative stress in different ways. Normally, hnRNPA1 binds to FEN1 rG4 and participates in the translational regulation of FEN1. Our results showed that hnRNPA1 knockdown decreased the expression of FEN1 and increased the level of γH2AX, which may be due to the DNA replication stress caused by the downregulation of FEN1. Under oxidative stress, the rG4 structure is disrupted and its binding to hnRNPA1 is reduced, which increases mRNA instability and degradation. Although we do not understand the underlying mechanism by which hnRNPA1 interacts with rG4 and maintains mRNA stability, our results demonstrate the importance of rG4 and its interaction with hnRNPA1 for FEN1 translation under oxidative stress.

## 5. Conclusions

In conclusion, this work provides new insights into the regulation of FEN1 expression under oxidative stress by the association between rG4 structure in FEN1 5′UTR and hnRNPA1. This study complements the existing research on FEN1 expression regulation and provides a novel model of cellular responses to oxidative stress. It also reveals the importance of the RNA secondary structural changes for transcript stability and translation and expands our understanding of gene expression pattern under different conditions. This work highlights the oxidation of G in the rG4-forming sequence and its epigenetic potential to affect gene expression. An exploration of rG4 self-regulation will assist in revealing novel mechanisms of gene expression regulation associated with diseases.

## Figures and Tables

**Figure 1 antioxidants-12-00276-f001:**
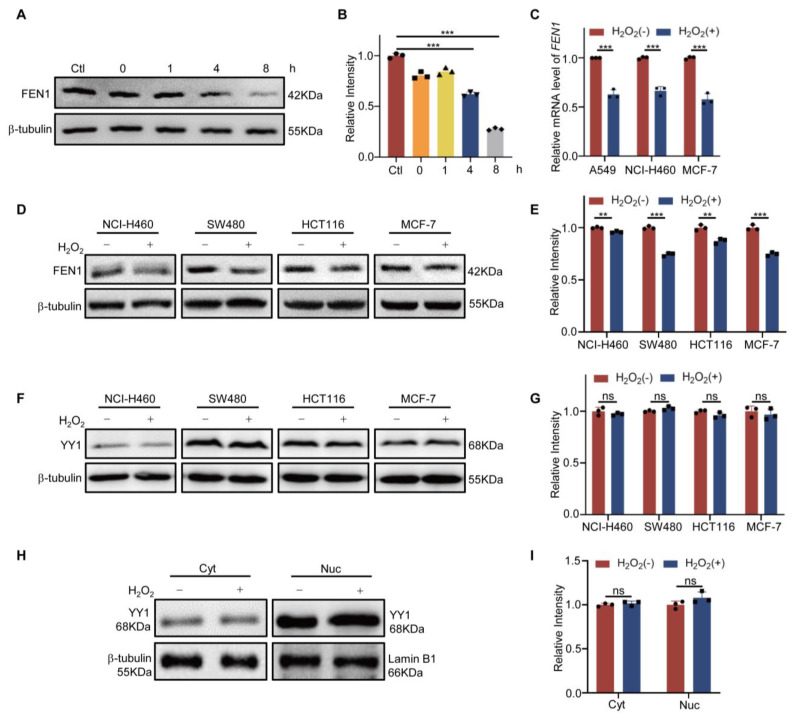
FEN1 is downregulated under oxidative stress. (**A**) Western blot detected the FEN1 protein level of MCF−7 cells recovered at different time points after 300 μM H_2_O_2_ treatment for 2 h. β−tubulin was used as an internal control. (**B**) Quantification of the marks shown in the panel (**A**). (**C**) A549, NCI−H460 and MCF−7 cells were treated with 300 μM H_2_O_2_ for 2 h and then recovered for 4 h. RT−qPCR was used to measure FEN1 mRNA level and β−actin was used as a loading control. (**D**) NCI−H460, HCT116, MCF−7 and SW480 cells were treated as in (**C**). Western blot was used to measure FEN1 protein levels and β−tubulin was used as a loading control. (**E**) Quantification of the marks shown in the panel (**D**). (**F**) After the cells were treated as in panel (**C**), the protein level of YY1 was detected by Western blot. (**G**) Quantification of YY1 levels in (**F**). (**H**) Nucleocytoplasmic separation assay was used to detect the intracellular distribution of YY1 under oxidative stress. β−tubulin was used as an internal reference for cytoplasm, and Lamin B1 was used as an internal reference for the nucleus. (**I**) Quantitative analysis of panel (**H**). The data represent the mean ± SD of three independent experiments. Student’s *t*-test, ** *p* < 0.01; *** *p* < 0.001; ns, no significance.

**Figure 2 antioxidants-12-00276-f002:**
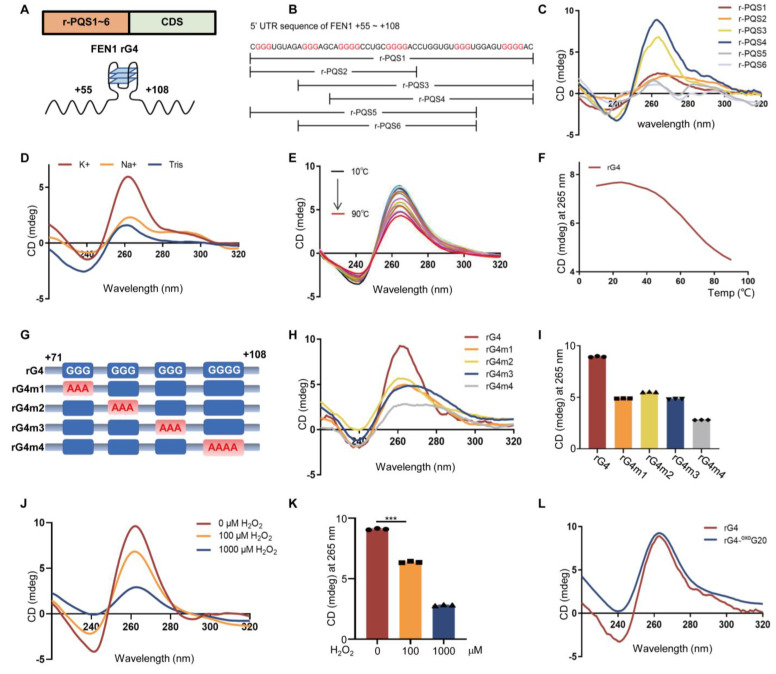
**A** G−quadruplex structure exists in the 5′UTR region of FEN1 mRNA. (**A**) Description of the G4 position in FEN1 mRNA. (**B**) There are six G4−forming sequences (r−PQS1~6) in the 5′UTR region of FEN1 mRNA. (**C**) A total of 5 μM PQS was folded into the G4 structure in G4 buffer and then detected by CD respectively. (**D**) Effect of metal ions on the CD spectra of G−quadruplexes. The effect of KCl (red line) and NaCl (orange line) on the ellipticity signal compared to the signal without salt. All CD data were obtained at 25 °C with an oligonucleotide concentration of 5 μM. (**E**) The effect of temperature (10~90 °C) on the CD spectrum of FEN1 rG4. At 90 °C, rG4 still produced a strong CD spectral signal of the G−quadruplex. (**F**) Temperature dependence of FEN1 rG4 ellipticity at 265 nm. (**G**) The effect of different G−tracks on rG4 formation. We replaced guanine (G) with adenine (A) in each G−track, and obtained 4 mutants, labeled as rG4m1~4. (**H**) The oligonucleotides containing wild type or mutations of rG4 (rG4m1~4); the samples (5 μM) were used for CD spectra. (**I**) Molar ellipticity at 265 nm of individual mutants. (**J**) CD spectra for FEN1 5′UTR RNA oligonucleotides, which were folded in G4 buffer to form an rG4 structure, with or without different concentrations of H_2_O_2_. (**K**) Statistical analysis of ellipticity at 265 nm for each sample. (**L**) The key site of the FEN1 rG4−forming sequence was labeled with an 8−oxoG and detected by CD after annealing. Data shown are the mean ± SD from three independent experiments. *** *p* < 0.001 as determined by the Student’s *t*-test.

**Figure 3 antioxidants-12-00276-f003:**
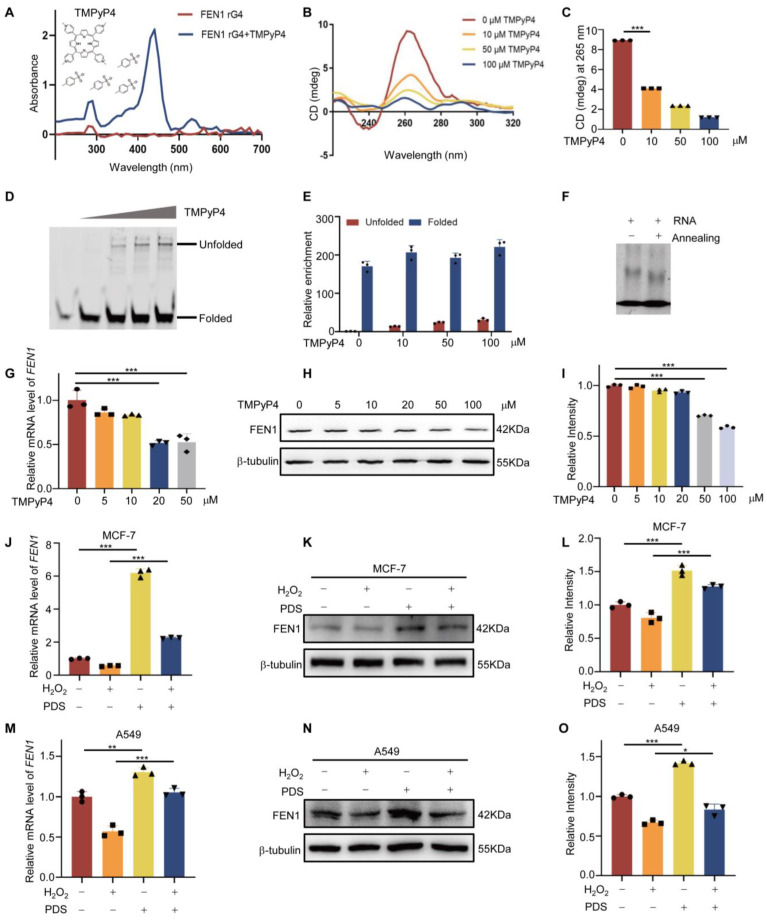
TMPyP4 affects rG4 formation. (**A**) Visible absorption spectra of pre−folded FEN1 rG4 in the absence or presence of TMPyP4. A total of 100 mM of TMPyP4 was added to 5 μM of the pre−folded FEN1 rG4, then we detected the absorbance value of 200~700 nm. (**B**) CD spectra of 5 μM pre−folded FEN1 rG4 (in 100 mM KCl) in the absence and presence of increasing concentrations of TMPyP4. (**C**) Statistical analysis of ellipticity at 265 nm for each sample. (**D**) The effect of TMPyP4 on the migration pattern of FEN1 rG4. FAM−labeled RNA or rG4 formed after annealing with or without TMPyP4, and migrated in 12% native−PAGE gel assay. (**E**) Statistical analysis of rG4 folding degree. (**F**) The migration of the annealed or unannealed FAM−labeled RNA or rG4 in the gel was detected by EMSA. (**G**) MCF−7 cells were treated with different concentrations of TMPyP4 for 12 h. RT−qPCR was used to measure the FEN1 mRNA level and β−actin was used as a loading control. (**H**) Western blot detection of protein changes of FEN1 under different concentrations of TMPyP4 treatment. (**I**) Changes in FEN1 protein concentration by quantitative analysis. (**J**) MCF−7 cells were treated with 300 μM H_2_O_2_ for 2 h and then recovered for 4 h with or without 10 μM pyridostatin (PDS) treatment. The mRNA level of FEN1 was measured by RT−qPCR. (**K**) Cells were treated as in (**J**). Western blot was used to measure FEN1 protein levels. (**L**) Quantitative analysis of panel (**K**). (**M**) A549 cells were treated as in (**J**), and RT−qPCR was used to measure the FEN1 mRNA level. (**N**) A549 cells were treated as in (**J**); the protein level of FEN1 was measured by Western blot. (**O**) Quantification of FEN1 levels in (**N**). Data are shown as mean ± SEM, n = 3. * *p* < 0.05; ** *p* < 0.01; *** *p* < 0.001, Student’s *t*-test.

**Figure 4 antioxidants-12-00276-f004:**
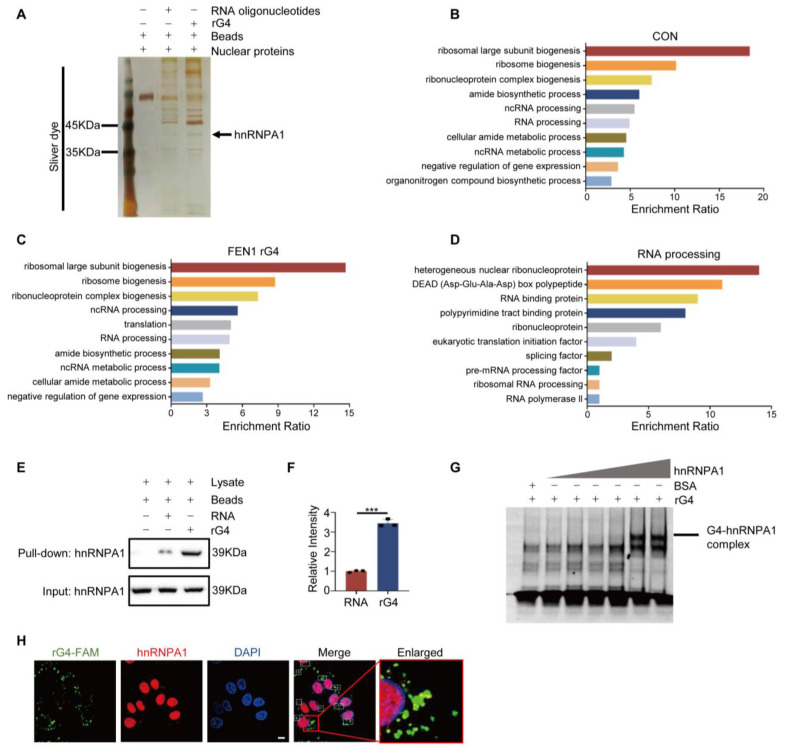
FEN1 rG4 interacts with hnRNPA1. (**A**) FEN1 rG4 binds to hnRNPA1. MCF−7 nuclear extracts were incubated with biotin−labeled oligonucleotides bound to M280 magnetic beads, rG4 or magnetic beads alone. SDS−PAGE for bound proteins and silver staining for total proteins. (**B**,**C**) GO analysis was performed on the LS−MS/MS results. Analysis of biological processes in control (**B**) and FEN1 rG4 (**C**) groups. (**D**) RNA processing−related proteins were classified by their function. (**E**) Pull−down experiments with hnRNPA1 antibody. (**F**) Quantification of hnRNPA1 levels in (**E**). (**G**) The annealed RNA oligonucleotides were incubated with different concentrations of hnRNPA1, and then the samples were analyzed on a 12% native PAGE gel. The upper band is the complex of hnRNPA1 and rG4. (**H**) Representative confocal fluorescence micrographs of FAM−tagged rG4 (green) and hnRNPA1 (red) in MCF−7 cells. Nuclei were stained with DAPI (blue). The colocalized rG4/hnRNPA1 foci are indicated by white boxes. Scale bar: 5 μm. The data represent the mean ± SD of three independent experiments. *** *p* < 0.001, Student’s *t*-test.

**Figure 5 antioxidants-12-00276-f005:**
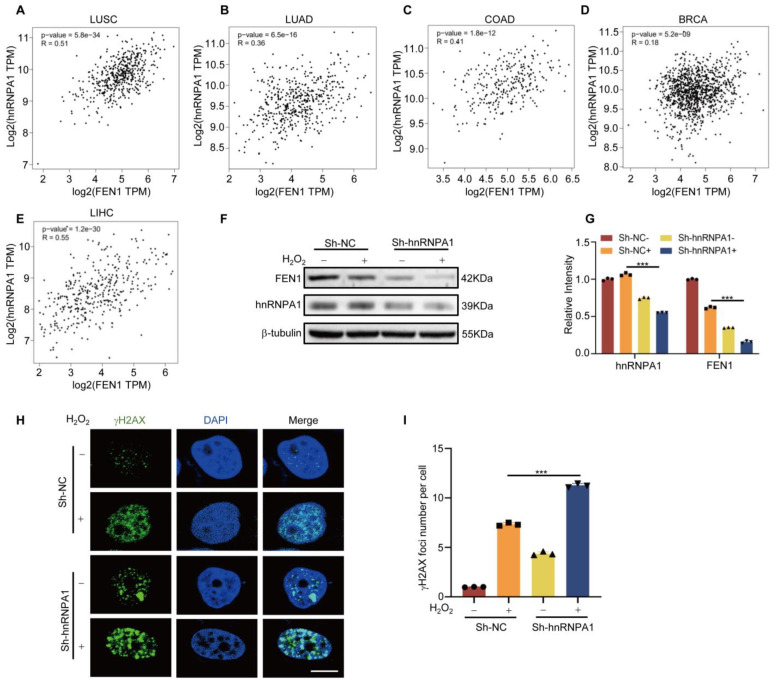
HnRNPA1 regulates FEN1 expression. (**A**–**E**) The correlation analysis of FEN1 and hnRNPA1 expression was used by the GEPIA database. (**F**) After knockdown of hnRNPA1 in MCF−7 cells, the protein level of FEN1 was detected by Western blot. (**G**) Quantitative analysis of changes in the protein level of FEN1. (**H**) After treatment as shown in (**F**), the levels of γH2AX were detected by immunofluorescence. γH2AX was labeled with FITC (green) and nuclei were labeled with DAPI (blue). Merged images show overlap between nuclei and γH2AX. Scale bar: 5 μm. (**I**) Quantitative analysis of fluorescence intensity of γH2AX. Data shown are means ± SD from three independent experiments. Student’s *t*-test, *** *p* < 0.001.

## Data Availability

All data are included in the article and Appendix A.

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
