# Peer review of "RNA G-Quadruplex within the 5′-UTR of FEN1 Regulates mRNA Stability under Oxidative Stress"

_antioxidants, 2023, doi:10.3390/antiox12020276_

Round 1
Reviewer 1 Report
The manuscript starts with the observation that FEN1 protein levels are strongly reduced upon H2O2 treatment in different cancer cell lines. This occurs independently of YY levels, what pushed the authors to look for alternative regulation factors. Bioinformatic analysis of the 5’UTR region of FEN1 gene reveals the existence of potential G4 forming sequences, among which number 4 is selected and confirmed. A mutant in this sequence is also crated for further experiments, but it is not used later on in vivo. Further in vitro experiments demonstrate that H2O2 treatment of the existence of 8oxoG instead of G affects the stability of the G4 structure. G4 destabilization by TMPyP4 decreased FEN1 protein levels. A search for proteins that interact with this G4 identifies several RNA binding factors, among which hnRNPA1 was confirmed. Up to here, the manuscript is well conducted and nice. However, hnRNPA1 knockdown still allows FEN1 downregulation upon H2O2 treatment (Figure 5B) and the effect on DNA damage seems independent on oxidative stress.
Thus, in general, the manuscript could be interesting at the level of FEN1 expression regulation but the consequences of the loss of this regulation or the mechanism need further investigation
Specific comments that need to be addressed:
- The manuscript starts with the observation that FEN1 protein levels are strongly reduced upon H2O2 treatment in different cancer cell lines. However, it has been previously reported that FEN1 levels decrease in HeLa cells upon oxidative damage. Have the authors tested HeLa cells themselves? Can they reproduce published data?
-the authors claim a regulation at the level of mRNA since the model is that the mRNA stability is reduced but the current version of the manuscript show only western blots. Northern blots and/or RT-qPCR experiments are required.
-why is rPQS3 not selected? The profile in Figure 2C is quite similar to rPQS4
-what would be the effect of G4 stabilization by pyridostatin? Are the authors able to detect increased FEN1 expression upon oxidative stress in this case?
-figure 5A. the correlation coefficient is rather low and I am not sure that this adds much to the paper at this point. Moreover, why is only breast cancer analyzed and not other tumors?
-figure 5E, isn’t significant the increase of gH2AX foci significant in sh hnRNPA1 cells (untreated with H2O2)? Has this been reported before? This should be discussed. In other words, the increase of damage seems independent on H2O2.
Other minor comments:
-the statistic used in each panel is not clear or indicated in the figure legends.
-Redundancy in line 51: In the BER system...in BER.
-line 55: word missing?: hypomethylation of FEN1 promoter?
-line 63: word missing?: the key role of FEN1
-line 315: n missing: knockdown
Reviewer 2 Report
Brief summary. This article describes the regulation of FEN1 (Flap endonuclease 1) expression under oxidative stress during DNA base excision repair (BER) by a RNA G-quadruplex (rG4) within its mRNA 5’ untranslated region (UTR). The important of this RNA secondary structure and stability is highlighted. The authors propose a model of regulation based on the interaction of hnRNPA1 (heterogeneous nuclear ribonucleoportein in RNA binding protein) to FEN1 rG4. They have used a combination of complementary methods, from cellular assays to biochemistry and biophysics measurements to validate their findings.
In the discussion, an explanation of the possible mechanism of hnRNAPA1 – FEN1 interaction under oxidative stress is missing and it would be helpful.
Specific comments:
In a few figures, the characters are too small to read: Fig. 2B, Fig. 3A, Fig. 4B and 4C
Fig. 1E. Downregulation effects not clear in this western blot. Is it significant?
Fig. 3G. FEN1 protein changes under different concentration of TMPyP4 not clear in this western blot. Is it significant?
It would be more comforting to use an orthogonal method to validate the direct interaction of FEN1 rG4 with hnRNPA1 in solution.
